# Home-Based Nonoperative-Side Quadriceps Neuromuscular Electrical Stimulation Prevents Muscle Weakness Following Anterior Cruciate Ligament Reconstruction

**DOI:** 10.3390/jcm11020466

**Published:** 2022-01-17

**Authors:** Vanessa Wellauer, Julia F. Item, Mario Bizzini, Nicola A. Maffiuletti

**Affiliations:** 1Human Performance Lab., Schulthess Clinic, 8008 Zurich, Switzerland; vanessa.wellauer@kws.ch (V.W.); julia.item@kws.ch (J.F.I.); mario.bizzini@kws.ch (M.B.); 2Performance Diagnostics, Schulthess Clinic, 8008 Zurich, Switzerland

**Keywords:** knee surgery, anterior cruciate ligament, quadriceps muscle, strength, activation

## Abstract

We compared the effectiveness of a home-based neuromuscular electrical stimulation (NMES) program applied to the quadriceps of the nonoperative side against sham-NMES as a complement to standard rehabilitation on knee extensor neuromuscular function in patients following anterior cruciate ligament (ACL) reconstruction. Twenty-four patients completed the 6 week NMES (*n* = 12) and sham-NMES (*n* = 12) post-operative interventions and were tested at different time points for neuromuscular function and self-reported knee function. Isometric, concentric, and eccentric strength deficits (muscle weakness) increased significantly from pre-surgery to 24 weeks post-surgery in the sham-NMES group (*p* < 0.05), while no significant changes were observed in the NMES group. On the stimulated (nonoperative) side, quadriceps voluntary activation and muscle thickness were respectively maintained (*p* > 0.05) and increased (*p* < 0.001) as a result of the NMES intervention, contrary to sham-NMES. Self-reported knee function improved progressively during the post-operative phase (*p* < 0.05), with no difference between the two groups. Compared to a sham-NMES intervention, a 6 week home-based NMES program applied to the quadriceps of the nonoperative side early after ACL reconstruction prevented the occurrence of knee extensor muscle weakness 6 months after surgery. We conclude that nonoperative-side NMES may help counteract muscle weakness after ACL reconstruction.

## 1. Introduction

In healthy subjects, unilateral resistance training of a single muscle group results in significant strength gains of the trained muscle but also—to a lesser extent—of the non-trained homonymous muscle on the contralateral side [1,2]. This adaptive phenomenon, which is commonly attributed to neurally mediated mechanisms and is usually referred to as cross-education [1,2,3], has been observed for a variety of muscles (knee extensors, ankle plantar/dorsiflexors, elbow flexors) and resistance training modalities (concentric, eccentric, isometric) including neuromuscular electrical stimulation (NMES) [1,2,4,5]. The value of cross-education for restoring muscle (and physical) function in patients with unilateral impairments induced by injury, surgery, or neurological damage has only recently been considered [6,7], and thus evidence for effectiveness is still limited, and clinical utility remains elusive.

For example, considerable deficits in knee extensor muscle strength (i.e., muscle weakness) persist for years following anterior cruciate ligament (ACL) reconstruction, with inevitable functional consequences [8,9]. Because the impossibility to adequately exercise the knee extensor muscles on the operative side early after ACL reconstruction surgery seems to be one of the main contributors to the resulting muscle weakness, contralateral (i.e., nonoperative-side) resistance training has been proposed as a potential countermeasure against muscle weakness [10,11,12,13]. Even if this type of training is not considered in ACL rehabilitation guidelines [14], in the last 10 years, four randomized controlled trials have investigated the effectiveness of different contralateral training protocols on knee extensor muscle strength early following ACL reconstruction [10,11,12,13]. In all these studies, resistance exercise consisted of concentric and/or eccentric voluntary contractions of the knee extensors of the nonoperative side performed on variable-resistance weight-lifting (leg extension and leg press) or isokinetic apparatuses, always as a complement to standard rehabilitation. The observed cross-education effect had positive short-term and/or long-term effects on operative-side knee extensor strength in all the studies, except in the one entailing the lowest contraction intensities [13]. Interestingly, higher-intensity contractions involving high-threshold fast motor unit recruitment have recently been suggested to play a role in explaining cross-education occurrence and magnitude [11]. In this perspective, it would be extremely relevant to explore the cross-education effect induced by NMES, which has the unique ability to recruit fast (in addition to slow) motor units even at relatively low contraction intensities [15].

Therefore, we conducted a randomized controlled trial in which we compared the effectiveness of a home-based contralateral NMES program against sham-NMES as a complement to standard rehabilitation on knee extensor neuromuscular function in patients following ACL reconstruction. On the basis of a previous study conducted with healthy subjects [4], we hypothesized that NMES would be particularly effective to improve knee extensor strength and therefore to counter muscle weakness on the operative side.

## 2. Materials and Methods

### 2.1. Patients and Study Design

A total of 30 patients scheduled for ACL reconstruction surgery at our institution were initially enrolled in the study (Figure 1). The main inclusion criteria were age between 18 and 50 years, unilateral ACL tear with/without meniscus and cartilage damage, and time between ACL injury and surgery <2 years. The main exclusion criteria were history of a lower limb injury on the contralateral side that required surgery, previous/current cardiorespiratory or neurological conditions, and pregnancy. Three patients declined to participate after having been informed about the study procedures; therefore, 27 patients provided written informed consent and were randomly assigned (1:1) to receive NMES (*n* = 13) or sham-NMES (*n* = 14). These patients underwent arthroscopically-assisted ACL reconstruction surgery using the hamstring or patellar tendon graft [16], completed a 24 week standard rehabilitation program according to the guidelines of our clinic [17], completed a 6 week intervention phase with either NMES or sham-NMES applied to the nonoperative side, and were tested for neuromuscular function and self-reported knee function 2 weeks before surgery (pre) and on post-operative week 2 (beginning of intervention), week 8 (end of intervention), and week 24 (end of rehabilitation/study) (Figure 2). Two patients from the sham-NMES group discontinued the intervention and one patient from the NMES group was excluded from the analyses due to improper NMES use (NMES applied to both sides), resulting in a sample size of 12 patients per group. Their general characteristics are presented in Table 1. The study protocol was approved by the local ethics committee and was registered at ClinicalTrials.gov (NCT01901965).

### 2.2. Standard Rehabilitation Program

The rehabilitation program lasted 24 weeks and included a maximum of 36 supervised physical therapy sessions. The program, which followed the post-operative guidelines provided by our institution [16,17], was either conducted in the physical therapy department of our clinic or in external rehabilitation facilities. During the very early post-operative period (≈3 days in hospital), the main focus was on knee range of motion, muscle control (knee extensors), and management of pain and swelling. After hospital discharge, the program mainly consisted of strengthening, proprioception, coordination, and cardiovascular exercises whose volume and intensity were progressively increased. Strengthening exercises targeted different lower extremity muscles (knee extensors, knee flexors, hip and calf muscles). For each exercise, patients were asked to complete three sets of 12–20 repetitions with an intensity of 65–80% of one repetition maximum. During the first 4–6 weeks after surgery, knee flexion was limited to 60° for weight-bearing exercises such as squats and lunges. Proprioception and coordination exercises consisted of moderate-intensity tasks with a focus on neuromuscular control of the operative knee such as stabilization drills on stable/unstable surfaces and single-leg balance (≈10 min per session). Cardiovascular exercise consisted of light-to-moderate intensity cycling on an ergometer (≈10 min per session).

### 2.3. Interventions (NMES and Sham-NMES)

Both intervention programs (NMES and sham-NMES) were completed from post-operative week 2 to week 8, for a total duration of 6 weeks. For both programs, the first session was completed in the clinic under the supervision of a qualified physical trainer; all the other sessions were conducted at home (18 sessions in total) with the possibility to contact the trainer by phone/e-mail on each session day. The intervention programs consisted of 3 weekly sessions of 20 min of NMES or sham-NMES that were applied to the quadriceps of the nonoperative side by using of a garment-based device (Kneehab XP, Bio-Medical Research Ltd., Galway, Ireland). The stimulation unit delivers spatially distributed current pathways (frequency: 50 Hz; pulse duration: 150–400 µs; on-off time: 5–10 s with a ramp up of 1 s and a ramp down of 0.5 s) by means of four large electrodes that are integrated into a knee/thigh brace. This solution has recently been shown to be more effective than conventional NMES (with no spatial distribution) for restoring knee extensor strength and physical function following ACL reconstruction surgery [18]. Patients were asked to apply by themselves NMES and sham-NMES in a comfortable seated position (e.g., on a standard chair), with the knee fixed at 90° by means of a strap. In the NMES group, patients were consistently asked to increase the stimulation intensity through each session to a maximally tolerated level, which resulted in visible, sustained, tetanic quadriceps contractions with superior patellar glide [19]. Patients were instructed on the importance of reaching the highest possible muscle tension during each stimulation train [20]. In the sham-NMES group, patients were asked to increase the stimulation intensity to a level where current could be perceived but with no resultant visible contraction of the quadriceps muscle [21]. Patients from both groups were not blinded to the type of NMES intervention they received.

### 2.4. Assessments

The primary outcomes were knee extension strength (isometric, concentric and eccentric) with respective strength deficits (estimates of muscle weakness) that were evaluated at all time points for the nonoperative side vs. at pre and week 24 for the operative side. The secondary outcomes were quadriceps voluntary activation and muscle thickness (estimate of muscle mass) that were evaluated at all time points for the nonoperative side vs. at pre and week 24 for the operative side, as well as self-reported knee function at all time points.

### 2.5. Knee Extensor Strength

Maximal voluntary isometric, concentric, and eccentric strength of the knee extensors were evaluated using an isokinetic dynamometer (Biodex Medical Systems, Shirley, NY, USA). Patients were comfortably seated (90° between the trunk and the thigh) and stabilized to the chair of the dynamometer with pairs of straps across the shoulders and the abdomen. The lever arm of the dynamometer was attached to the leg ≈5 cm proximal to the lateral malleolus with a strap. The lateral femoral condyle was consistently aligned with the axis of the dynamometer, and gravity-correction procedures were systematically followed prior to testing. For the evaluation of isometric strength, the knee was fixed at 90° and subjects were asked to maximally contract their knee extensor muscles during ≈5 s, with a progressive force build up during the first 1–2 s of the contraction. For the evaluation of concentric and eccentric strength, angular velocity was 60°/s and knee extension/flexion range of motion was 80°, from 90° to 10° of knee flexion. Subjects were consistently instructed to extend the knee as strong as possible during the concentric phase, as well as to offer the maximal resistance to the lever through the entire range of motion during the eccentric phase. For each contraction mode, subjects completed four quasi-maximal familiarization trials, followed by three consecutive maximal trials. The nonoperative knee was always tested first, followed by the operative one. Rest periods of 3 and 5 min were interspersed between contraction modes and sides, respectively. Isometric, concentric, and eccentric strength of the operative and nonoperative side was quantified as the highest peak torque recorded during the different trials. Knee extension isometric, concentric, and eccentric strength deficits were calculated using the following formula: (nonoperative-side strength—operative-side strength)/nonoperative-side strength × 100 [22].

### 2.6. Quadriceps Voluntary Activation and Muscle Thickness

Voluntary activation was evaluated by delivering electrical stimuli to the quadriceps muscle belly both during (≈2 s from contraction onset) and ≈2 s after the maximal isometric contractions, according to the methodology described by Wellauer et al. [23]. Briefly, paired stimuli with an interpulse interval of 10 ms and an intensity of 100 mA were delivered with a high-voltage (maximal voltage: 400 V), constant-current stimulator (Model DS7AH, Digitimer Ltd., Hertfordshire, United Kingdom) connected to four large electrodes integrated into a brace (Kneehab XP, Bio-Medical Research Ltd., Galway, Ireland). Paired stimuli delivered during and after the contraction evoked a superimposed and a potentiated twitch, respectively. Quadriceps voluntary activation was subsequently calculated with this formula: [100 − (superimposed twitch torque/potentiated twitch torque) × 100] (mean of three trials).

Muscle thickness was evaluated with B-mode ultrasonography (MyLab 25, Esaote, Florence, Italy), according to the methodology described by Casartelli et al. [24]. Briefly, a linear array probe (frequency: 10–15 MHz) was positioned longitudinally over the lateral aspect of the vastus lateralis muscle (50% of femur length) while patients were comfortably seated and relaxed with the knee at 90° of flexion. Three images (width: 3.8 cm; depth: 4–6 cm) were recorded per side and patient. Subsequently, muscle thickness was measured with an image-editing software (ImageJ 1.36b, National Institute of Health, Bethesda, MD, USA) as the distance between the superficial and deep aponeurosis of the vastus lateralis muscle on three image spots (left, middle, right). For each side and patient, muscle thickness was retained as the mean value of the three spots and images.

### 2.7. Self-Reported Knee Function

Self-reported knee function was evaluated with the Knee Injury and Osteoarthritis Outcome Score (KOOS) questionnaire [25], which measures patients’ opinions about their knee and associated problems, and is often used after ACL reconstruction [26]. There are 42 questions in the questionnaire, which are further categorized into five subscales: pain, symptoms, activities of daily living, sport, and quality of life. There are five possible answers to each question, and each question is subsequently scored from 0 (highest impairment) to 4 (no impairment). For each subscale, except sport, scores were added to produce a single subscale score that was converted to values ranging from 0 (highest impairment) to 100 (no impairment).

### 2.8. Statistics

On the basis of previously documented increases in contralateral maximal eccentric strength following NMES training (+34%) vs. no training (+4%) [4], we performed an a priori power analysis with MedCalc (MedCalc statistical software, Ostend, Belgium) for which the required sample size necessary to obtain a significant difference in strength gains between NMES and sham-NMES was 13 patients per group (type I error: 0.05; type II error: 0.20).

Normality was checked for all the variables with Shapiro–Wilk tests. Between-group differences in general characteristics were examined with unpaired *t*-tests. For all primary and secondary outcomes, we used two-way (group by time) and three-way (group by side by time) ANOVAs with repeated measures on time and side followed by Fisher LSD post hoc comparisons. The level of significance was set at *p* < 0.05.

## 3. Results

### 3.1. Primary Outcomes

In both groups, nonoperative-side isometric strength decreased from pre to week 2 (*p* < 0.01), was unchanged from week 2 to week 8, and finally increased from week 8 to week 24 (*p* < 0.01), with no difference between week 24 and pre (Table 2). Nonoperative-side concentric strength increased from pre to week 24 (*p* < 0.001) and was higher at week 24 than at all the other time points (*p* < 0.001), but did not differ between pre, week 2, and week 8. Nonoperative-side eccentric strength increased from pre to week 8 (*p* < 0.05), with no difference between pre and week 2, and then increased again from week 2 to week 24 (*p* < 0.05), reaching a level higher than pre (*p* < 0.05). For all strength outcomes and in both groups, values were lower on the operative compared to the nonoperative side (*p* < 0.001), both at pre and at week 24.

Isometric (Figure 3A), concentric (Figure 3B), and eccentric (Figure 3C) strength deficits did not change significantly from pre to week 24 in the NMES group, while all deficits increased over time in the sham-NMES group (*p* < 0.05).

### 3.2. Secondary Outcomes

Quadriceps voluntary activation of both sides did not change significantly over time in the NMES group (Figure 4A). On the other hand, voluntary activation of the nonoperative side decreased from pre to week 8 (*p* < 0.01)—likely reflecting activation failure—and then reincreased from week 8 to week 24 (*p* < 0.05) in the sham-NMES group (Figure 4B), while no significant time-related changes were observed for the operative side. Quadriceps muscle thickness of the nonoperative side increased from pre to week 8 (*p* < 0.001) in the NMES group (Figure 4C)—likely reflecting muscle hypertrophy—but did not change significantly in the sham-NMES group (Figure 3D). In both groups, operative-side muscle thickness showed a significant decline from pre to week 8 (*p* < 0.001)—likely reflecting muscle atrophy.

In both groups, KOOS pain, symptoms and activities of daily living scores decreased from pre to week 2 (*p* < 0.01), increased from week 2 to week 8 (*p* < 0.001)—while being still lower than pre values (*p* < 0.05)—and finally increased again from week 8 to week 24 (*p* < 0.05), with no difference between week 24 and pre (Table 3). KOOS quality of life score decreased from pre to week 2 (*p* < 0.001), increased from week 2 to week 8 (*p* < 0.001), and also from week 8 to week 24 (*p* < 0.001), reaching a level higher than pre (*p* < 0.001).

## 4. Discussion

### 4.1. Main Findings

Compared to a sham-NMES intervention, a 6 week home-based NMES program applied to the quadriceps of the nonoperative side early after ACL reconstruction and as a complement to standard rehabilitation (1) prevented the occurrence of knee extensor muscle weakness in isometric, concentric, and eccentric conditions 6 months after surgery and (2) preserved muscle activation and increased muscle mass of the stimulated quadriceps (nonoperative side), but (3) was not superior to sham-NMES for improving patient-relevant outcomes such as symptoms, functional status, and health-related quality of life.

### 4.2. NMES Prevented Muscle Weakness 6 Months after Surgery

Isometric, concentric, and eccentric knee extensor strength deficits were preserved from pre-surgery to 6 months post-surgery in the NMES group but not in the sham-NMES group, for which a significant increase was observed. In other words, the NMES intervention—but not sham-NMES—prevented the occurrence of additional knee extensor muscle weakness following ACL reconstruction. In our study, strength deficits were expressed as a sort of asymmetry ratio between the operative and nonoperative sides [22]. Using a similar approach, one ACL study observed comparable results of post-operative contralateral strength training on isometric strength deficits [12]. In two other studies, voluntary strength training of the nonoperative quadriceps resulted in a significant cross-education effect (i.e., an increase in strength of the operative-side non-trained muscle) immediately after the 8 week training program [10,11], which was, however, not maintained at the 6 month post-operative follow-up [11]. We could not verify the occurrence and magnitude of cross-education induced by the 6 week NMES intervention per se, as operative-side maximal voluntary strength is only assessed starting from 12–16 post-operative weeks in our hospital. However, the fact that NMES was able to prevent further deteriorations of knee extension muscle strength at the longer-term follow-up, as well as in all contraction modalities, certainly represents the most important finding of our study.

The effects of contralateral strength training on ACL-reconstructed knee extension strength have already been examined in four randomized controlled trials [10,11,12,13]. In these studies, an 8–12 week intervention consisting of 2–5 weekly sessions of strength training exercises was introduced as a complement to standard rehabilitation 1–4 weeks after surgery. The main exercises consisted of concentric and/or eccentric voluntary contractions of the knee extensors of the nonoperative side performed on variable-resistance weight-lifting (leg extension and leg press) or isokinetic apparatuses, with 3–5 sets of 3–12 repetitions at different exercise intensities (70–100% max) depending on the study. Interestingly, the only study in which contralateral strength training had no positive effect on knee extensor muscle function was the one using concentric contractions with the lowest intensities (70–80% max in [13]). This led us and others [11] to conjecture that fast motor unit recruitment—which can be obtained with high-intensity voluntary contractions but also with NMES [15], such as in this study—may be involved in explaining the occurrence and magnitude of the cross-education effect. This reasoning is based on recent findings showing that cross activation of the motor cortex during unilateral contractions has been found to increase pari passu with contraction intensity [27], and that unilateral eccentric resistance training—particularly if involving high-intensity contractions—has been shown to induce large contralateral strength gains [1,28]. As far as NMES is concerned, this modality has the unique ability to recruit motor units with a non-selective pattern whereby fast (in addition to slow) motor units are recruited even at relatively low force levels [15]. This seems to be due to the fact that largest axons (innervating fast muscle fibers) are often located superficially in the quadriceps muscle [29] and thus closer to the stimulating electrodes. Therefore, we suppose that the preferential recruitment of fast motor units associated to the NMES intervention was at least partly responsible for the crossed-education effect and therefore for preventing the occurrence of knee extensor muscle weakness after ACL reconstruction.

Compared to conventional voluntary strength training, the two main advantages of NMES for cross-education purposes are that it does not require high-intensity contractions—which are not recommended soon after surgery—and it can easily be conducted at home with adequate instructions and minimal equipment.

### 4.3. NMES Preserved Activation and Increased Mass of the Stimulated Muscle

Our 6 week NMES training program resulted in classical neuromuscular adaptations on the stimulated quadriceps (nonoperative side). The disuse-related decline in muscle activation observed in the sham-NMES group at week 8 did not occur in the NMES group, therefore confirming that NMES has the capacity to prevent quadriceps activation failure that is common after ACL reconstruction surgery [30]. The NMES-induced increase in vastus lateralis muscle thickness at week 8 was not observed in the sham-NMES group, thereby confirming that NMES by itself can improve quadriceps muscle mass even during a period of disuse [31,32]. Surprisingly, knee extension strength of the stimulated muscle (nonoperative side) did not increase following the NMES intervention, which reinforces the recent contention that the effects of NMES on muscle mass could be more consistent and possibly precede those on muscle strength in the context of disuse [33]. On the other hand, our NMES intervention was unable to counter the decline of muscle mass on the non-stimulated quadriceps (operative side), which was consistently observed in both patient groups.

### 4.4. NMES Did Not Influence Self-Reported Function

Harput et al. [10] and Minshull et al. [11] reported no benefits of contralateral strength training on self-reported knee symptoms, function, and sports activities (International Knee Documentation Committee questionnaire) 6 months after ACL reconstruction, despite better muscle function at the short-term follow-up. These findings are fully in agreement with our study, even if we used another questionnaire (KOOS) that also captures activities of daily living and quality of life, which was administered at all time points, including immediately after the intervention. Besides trends and one previous study showing a group effect—but actually not a group by time interaction—for two cross-education training groups against a control group in knee functional status (Lysholm score; [34]), it seems that the preservation of knee extensor muscle strength provided by voluntary and NMES contralateral training occurs independently from subjectively-reported improvements in knee function. Similar findings have already been reported for the classical post-operative application of NMES to the operative side (not for cross education purposes) in ACL patients [18].

### 4.5. Limitations

Our study has several limitations worth noting. The sample size was quite small, which could have affected the statistical power of some results and also the generalizability of our current findings. It was actually pretty difficult to recruit patients for this study, mainly due to time constraints, and we even had to exclude a third group/intervention (home-based eccentric strength training) that could have provided a more realistic clinical comparison with respect to the NMES intervention. Another limitation is that NMES use was not fully controlled as we opted for a home-based administration of both interventions that is, however, known to be more convenient for patients, with increased comfort and privacy and decreased stress of travel. Moreover, we used a garment-based distributed NMES solution that was recently shown to achieve higher compliance rates and treatment effectiveness than conventional NMES (with no spatial distribution) for restoring knee extension strength and functional performance after ACL reconstruction [18]. Particular emphasis was placed on the importance of the “Fitzgerald” clinical criteria [19] in the NMES group (full, sustained, tetanic contraction of the quadriceps with superior patellar glide) [35], which guaranteed the presence of adequate muscle tension—i.e., the main prerequisite for NMES training effectiveness [20]—contrary to the sham-NMES group. Finally, self-report data should be interpreted with cation as patients were not blinded to the type of intervention they received.

## 5. Conclusions

We conclude that a 6 week home-based NMES program applied to the quadriceps of the nonoperative side early after ACL reconstruction and as a complement to standard rehabilitation is a potentially valuable countermeasure against post-operative knee extensor muscle weakness.

## Figures and Tables

**Figure 1 jcm-11-00466-f001:**
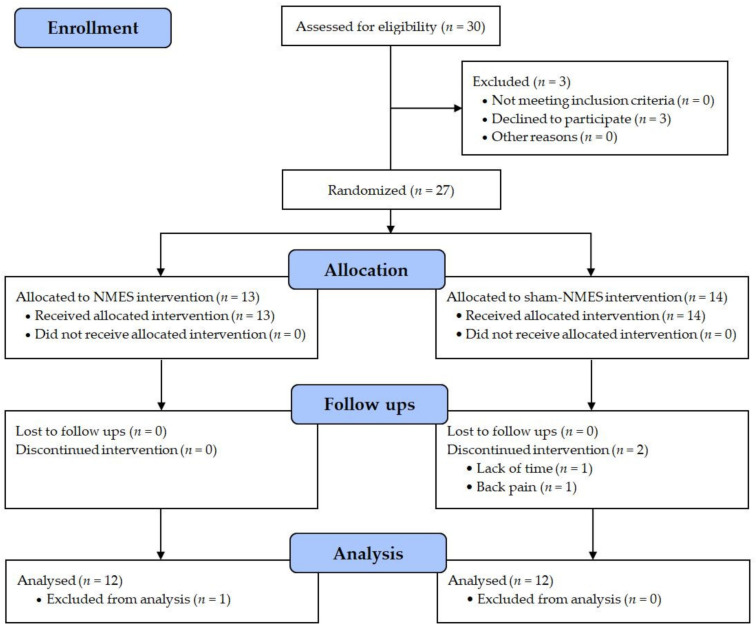
CONSORT flow diagram.

**Figure 2 jcm-11-00466-f002:**
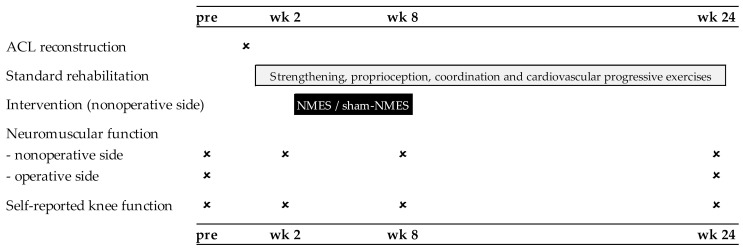
Overview of the experimental protocol.

**Figure 3 jcm-11-00466-f003:**
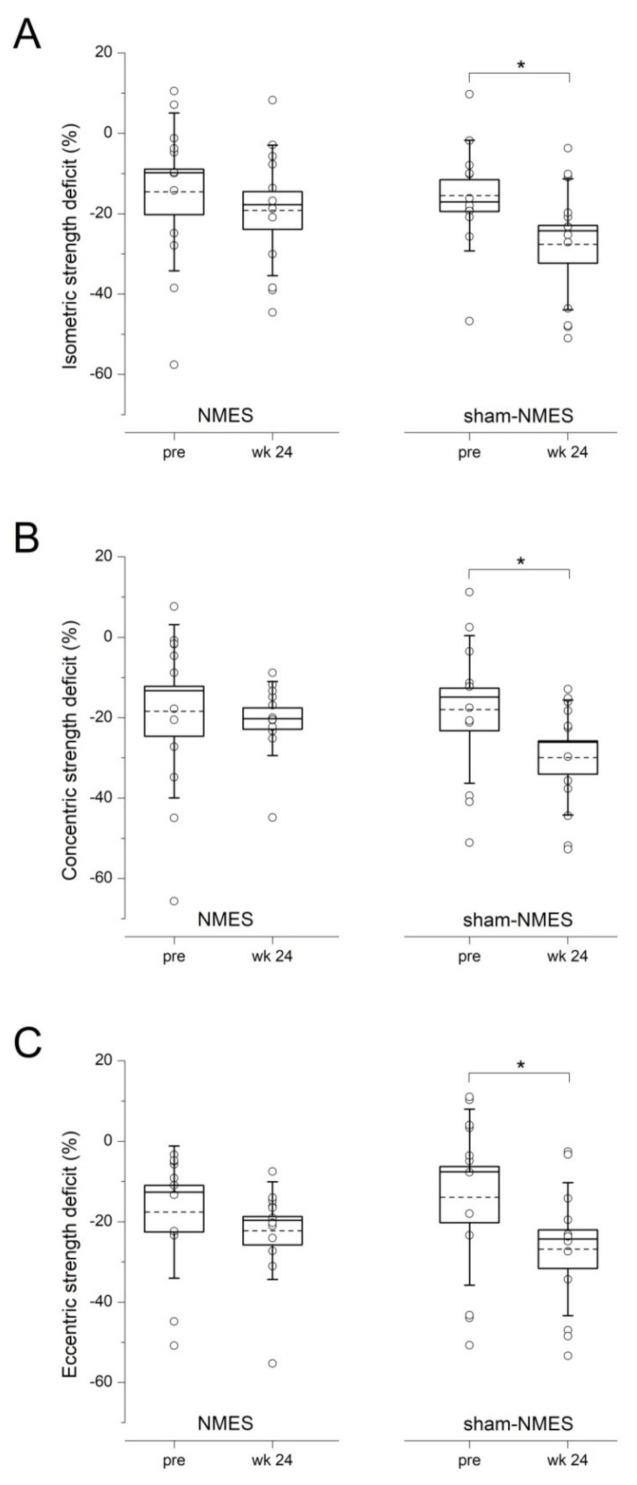
Isometric (**A**), concentric (**B**), and eccentric (**C**) knee extensor strength deficits at pre and week 24 in the NMES and sham-NMES groups. The box represents the standard error, the horizontal lines within the box represent the median and the mean, and the whiskers represent the standard deviation (single data are also shown). * *p* < 0.05.

**Figure 4 jcm-11-00466-f004:**
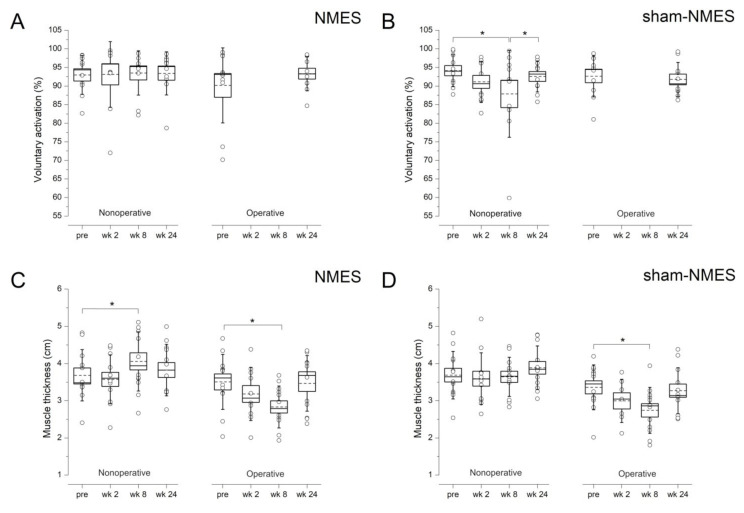
Quadriceps voluntary activation (**A**,**B**) and muscle thickness (**C**,**D**) of the operative and nonoperative side at the different time points in the NMES and sham-NMES groups. The box represents the standard error, the horizontal lines within the box represent the median and the mean, and the whiskers represent the standard deviation (single data are also shown). * *p* < 0.01.

**Table 1 jcm-11-00466-t001:** Patient characteristics by group.

	NMES (*n* = 12)	Sham-NMES (*n* = 12)	*p*-Value (*t*-Test)
Number of women/men	5/7	5/7	-
Age (years)	29 ± 6	30 ± 11	0.64
Height (cm)	172 ± 8	171 ± 6	0.56
Weight (kg)	68 ± 13	71 ± 14	0.56
Time from injury to surgery (months)	7 ± 8	8 ± 5	0.77
Number of HTG/PTG	11/1	9/3	-
Number of partial meniscectomies	6	5	-
Number of cartilage repairs	1	1	-
Preinjury Tegner score (0–10)	6.5 ± 1.7	7.2 ± 1.2	0.58
Preoperative Tegner score (0–10)	3.5 ± 1.7	3.6 ± 1.6	0.70

Mean data ± SD. HTG: hamstring tendon graft; PTG: patellar tendon graft.

**Table 2 jcm-11-00466-t002:** Knee extensor strength data by group, side, and time point.

	Group	Side	Pre	Week 2	Week 8	Week 24
Isometric strength	NMES	Nonoperative	216 ± 74	206 ± 72 ^a^	197 ± 72 ^a^	217 ± 77
(Nm)	NMES	Operative	188 ± 82 *	/	/	184 ± 70 *
	sham-NMES	Nonoperative	204 ± 53	186 ± 52 ^a^	195 ± 56 ^a^	212 ± 62
	sham-NMES	Operative	165 ± 56 *	/	/	155 ± 63 *
Concentric strength	NMES	Nonoperative	190 ± 55	188 ± 55	188 ± 56	209 ± 11 ^b^
(Nm)	NMES	Operative	158 ± 63 *	/	/	169 ± 57 *
	sham-NMES	Nonoperative	181 ± 49	181 ± 44	186 ± 45	199 ± 54 ^b^
	sham-NMES	Operative	148 ± 49 *	/	/	141 ± 55 *
Eccentric strength	NMES	Nonoperative	260 ± 77	260 ± 82	266 ± 78 ^c^	286 ± 86 ^d^
(Nm)	NMES	Operative	219 ± 88 *	/	/	224 ± 78 *
	sham-NMES	Nonoperative	232 ± 50	240 ± 55	253 ± 56 ^c^	244 ± 49 ^d^
	sham-NMES	Operative	198 ± 57 *	/	/	178 ± 53 *

Mean data ± SD. * Lower than nonoperative (*p* < 0.001). ^a^ Lower than pre and week 24 (*p* < 0.01). ^b^ Higher than all the other time points (*p* < 0.001). ^c^ Higher than pre (*p* < 0.05). ^d^ Higher than pre and week 2 (*p* < 0.05).

**Table 3 jcm-11-00466-t003:** Self-reported knee function (KOOS) data by group and time point.

	Group	Pre	Week 2	Week 8	Week 24
Pain (0–100)	NMES	88 ± 13	66 ± 17 ^a^	77 ± 16 ^bc^	87 ± 11
	sham-NMES	79 ± 14	66 ± 17 ^a^	73 ± 7 ^bc^	85 ± 9
Symptoms (0–100)	NMES	80 ± 14	54 ± 12 ^a^	72 ± 17 ^bc^	84 ± 13
	sham-NMES	74 ± 20	49 ± 13 ^a^	60 ± 17 ^bc^	79 ± 14
Activities daily living (0–100)	NMES	96 ± 5	65 ± 18 ^a^	90 ± 7 ^bc^	98 ± 3
	sham-NMES	93 ± 10	56 ± 16 ^a^	87 ± 9 ^bc^	97 ± 3
Quality of life (0–100)	NMES	39 ± 15 ^d^	20 ± 15 ^a^	41 ± 20 ^d^	54 ± 23
	sham-NMES	33 ± 16 ^d^	16 ± 12 ^a^	35 ± 18 ^d^	53 ± 22

Mean data ± SD. ^a^ Lower than all the other time points (*p* < 0.01). ^b^ Lower than pre and week 24 (*p* < 0.05). ^c^ Higher than week 2 (*p* < 0.001). ^d^ Lower than week 24 (*p* < 0.001).

## Data Availability

The data presented in this study are available on request from the corresponding author.

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
