# Peer review of "Home-Based Nonoperative-Side Quadriceps Neuromuscular Electrical Stimulation Prevents Muscle Weakness Following Anterior Cruciate Ligament Reconstruction"

_jcm, 2022, doi:10.3390/jcm11020466_

Round 1

Reviewer 1 Report

Thank you for asking me to review this manuscript. This is an important topic. The manuscript is generally well written and the conclusions are well-founded. There are a few things that would improve the script and make it easier to understand.
I would like to see a picture of the experimental setup - it would improve the understanding of the methods for the reader
The authors discuss the issues of recruitment of motor units, please discuss them in more detail and refer to the results obtained by this manuscript 
 Please show more evidence of Interventions (NMES and sham-NMES) to be more effective than conventional NMES (with no spatial distribution) for restoring quadriceps strength and physical function following ACL reconstruction surgery

Author Response

Dear reviewer, thank you very much for the valuable comments. We hope that the revisions will satisfy your standard. Please note that:

[C1.0] = comment;

[R1.0] = response;

{…} = action: changes in manuscript (Page, Section in the revised version).

[C1.1] Thank you for asking me to review this manuscript. This is an important topic. The manuscript is generally well written and the conclusions are well-founded. There are a few things that would improve the script and make it easier to understand.

[R1.1] Thank you.

[C1.2] I would like to see a picture of the experimental setup - it would improve the understanding of the methods for the reader.

[R1.2] We have added a new figure showing the experimental protocol. For the specific setup of all the assessments (knee extensor strength, quadriceps voluntary activation and muscle thickness) and NMES interventions, the reader can refer to our previously-conducted studies (that are cited in the respective sections of the Methods).

{Please see the new Figure 2}

[C1.3] The authors discuss the issues of recruitment of motor units, please discuss them in more detail and refer to the results obtained by this manuscript.

[R1.3] The Discussion section has been modified accordingly.

{Page 10; Section 4.2.: “As far as NMES is concerned, this modality has the unique ability to recruit motor units with a non-selective pattern whereby fast (in addition to slow) motor units are recruited even at relatively low force levels [15]. This seems to be due to the fact that largest axons (innervating fast muscle fibers) are often located superficially in the quadriceps muscle [29] and thus closer to the stimulating electrodes. Therefore, we suppose that the preferential recruitment of fast motor units associated to the NMES intervention was at least partly responsible for the crossed-education effect and therefore for preventing the occurrence of knee extensor muscle weakness after ACL reconstruction.”}

[C1.4] Please show more evidence of Interventions (NMES and sham-NMES) to be more effective than conventional NMES (with no spatial distribution) for restoring quadriceps strength and physical function following ACL reconstruction surgery.

[R1.4] We have addressed this comment in the limitations section.

{Page 11; Section 4.5.: “Also, we used a garment-based distributed NMES solution which was recently shown to achieve higher compliance rates and treatment effectiveness than conventional NMES (with no spatial distribution) for restoring knee extension strength and functional performance after ACL reconstruction [18].”}

Reviewer 2 Report

The authors report a electrical stimulation training for the contralateral quadriceps muscle after ACL reconstruction.

Introduction

What was the benefit in a home-based vs. hospital-based program?

Please conclude the introduction with one brief hypothesis.

Was the hypothesis that the contralateral or ipsilateral side improved?

Materials and Methods

Were the hamstring and patellar tendon graft used homogeneously among the two groups?

Why was one patient from the NMES group excluded?

Table 1: Please add the respective p-values.

How often did the participants practice at home vs. in the hospital on both groups?

It is important to mention that the patients in the sham-NMES group could not be blinded.

Assessments: The primary outcome was knee extension for which side?

Statistics: Which tests were used if the distribution was not normal?

Results

Primary outcomes: Why did the sham-NMES also improve from wk2 and wk8?

Secondary outcomes: Please explain in more detail why the muscle thickness in NMES is increased after 8 weeks in the uninvolved side.

Does the NMES really matter? In the end, voluntary activation and muscle thickness are not changed from pre to wk24.

Discussion

Do you have an explanation why your NMES intervention could not counter the muscle mass decline on the non-stimulated quad?

The self-reported questionnaire might not be useful due to the unblinded setting. This should also be mentioned in the limitations.

What is the rationale on a study-end at week 24?

Also, was the range-of-motion of the operatively treated knee improved?

Author Response

Dear reviewer, thank you very much for the valuable comments. We hope that the revisions will satisfy your standard. Please note that:

[C2.0] = comment;

[R2.0] = response;

{…} = action: changes in manuscript (Page, Section in the revised version).

[C2.1] The authors report a electrical stimulation training for the contralateral quadriceps muscle after ACL reconstruction.

[R2.1] No answer required.

Introduction

[C2.2] What was the benefit in a home-based vs. hospital-based program?

[R2.2] The benefits of home-based NMES – particularly from the patient perspective – have been presented in the limitations section.

{Page 10; Section 4.5.: “Another limitation is that NMES use was not fully controlled, as we opted for a home-based administration of both interventions which is however known to be more convenient for patients, with increased comfort and privacy and decreased stress of travel.”}

[C2.3] Please conclude the introduction with one brief hypothesis.

[R2.3] A brief hypothesis was already provided in the last sentence of the Introduction. This sentence has nevertheless been slightly modified to take into account the next comment (C2.4).

{Page 2; Section 1.: “Based on a previous study conducted with healthy subjects [4], we hypothesized that NMES would be particularly effective to improve knee extensor strength and therefore to counter muscle weakness on the operative side.”}

[C2.4] Was the hypothesis that the contralateral or ipsilateral side improved?

[R2.4] The hypothesis referred to the operative side and it has now been clarified (see previous reply). Thanks to this comment, we realized that the use of “contralateral” and “ipsilateral” with respect to the side was quite confusing. Therefore, we consistently used the terms “operative” and “nonoperative” when referring to the side, and we only used “contralateral” with reference to the training program.

Materials and Methods

[C2.5] Were the hamstring and patellar tendon graft used homogeneously among the two groups?

[R2.5] This information is provided in Table 1. The hamstring tendon graft was used in the vast majority of the patients from both groups (11 in the NMES group and 9 in the sham-NMES group).

[C2.6] Why was one patient from the NMES group excluded?

[R2.6] This information has been added.

{Page 2; Section 2.1.: “Two patients from the sham-NMES group discontinued the intervention and one patient from the NMES group was excluded from the analyses due to improper NMES use, resulting in a sample size of 12 patients per group.”}

[C2.7] Table 1: Please add the respective p-values.

[R2.7] Done accordingly.

{Please see Table 1}

[C2.8] How often did the participants practice at home vs. in the hospital on both groups?

[R2.8] This information is provided in the Interventions subsection.

{Page 4; Section 2.3.: “For both programs, the first session was completed in the clinic under the supervision of a qualified physical trainer; all the other sessions were conducted at home (18 sessions in total) with the possibility to contact the trainer by phone/e-mail on each session day.”}

[C2.9] It is important to mention that the patients in the sham-NMES group could not be blinded.

[R2.9] We have specified that all patients (from both groups) could not be blinded to the intervention.

{Page 4; Section 2.3.: “Patients from both groups were not blinded to the type of NMES intervention they received.”}

[C2.10] Assessments: The primary outcome was knee extension for which side?

[R2.10] Knee extension strength values of both sides (operative and nonoperative) – but evaluated at different time points – were used to calculate strength deficits. The text in the Assessments subsection has been slightly modified to address this comment.

{Page 5; Section 2.4.: “The primary outcomes were knee extension strength (isometric, concentric and ec-centric) with respective strength deficits (estimates of muscle weakness) that were evaluated at all time points for the nonoperative side vs at pre and wk 24 for the operative side.”}

[C2.11] Statistics: Which tests were used if the distribution was not normal?

[R2.11] Out of all the primary and secondary outcomes, only the data of one subscale (symptoms) of the self-reported questionnaire (KOOS) were not normally distributed. Therefore, also in order to avoid confusion due to multiple statistical procedures we preferred to make consistent use of the ANOVA. Of note, nonparametric testing provided exactly the same results as the two-way ANOVA for the subscale symptoms.

Results

[C2.12] Primary outcomes: Why did the sham-NMES also improve from wk2 and wk8?

[R2.12] None of the primary outcomes showed significant changes from wk 2 to wk 8 in the sham-NMES group.

[C2.13] Secondary outcomes: Please explain in more detail why the muscle thickness in NMES is increased after 8 weeks in the uninvolved side.

[R2.13] This was likely reflecting training-induced muscle hypertrophy (and has been acknowledged as such).

{Page 13; Section 3.2.: “Quadriceps muscle thickness of the nonoperative side increased from pre to wk 8 (p < 0.001) in the NMES group (Figure 4C) – likely reflecting muscle hypertrophy – but did not change significantly in the sham-NMES group (Figure 3D).”}

[C2.14] Does the NMES really matter? In the end, voluntary activation and muscle thickness are not changed from pre to wk24.

[R2.14] We understand this comment, and believe that the study was probably adequately powered for the primary outcomes, but not necessarily for the secondary and mechanistic outcomes of the operative side such as activation and thickness.

Discussion

[C2.15] Do you have an explanation why your NMES intervention could not counter the muscle mass decline on the non-stimulated quad?

[R2.15] We did not really expect a preservation of skeletal muscle mass for the non-stimulated quadriceps (operative side) in the absence of training/contractile activity.

[C2.16] The self-reported questionnaire might not be useful due to the unblinded setting. This should also be mentioned in the limitations.

[R2.16] This limitation has been acknowledged.

{Page 11; Section 4.5.: “Finally, self-report data should be interpreted with cation as patients were not blinded to the type of intervention they received.”}

[C2.17] What is the rationale on a study-end at week 24?

[R2.17] This corresponds to the end of the standard rehabilitation program for all our patients, beyond which it would have been extremely difficult to keep them involved in the study. Also, the same time period was considered in two similar studies (Harput et al. 2018; Minshull et al. 2021), which facilitated inter-study comparisons at similar time points.

[C2.18] Also, was the range-of-motion of the operatively treated knee improved?

[R2.18] Knee range of motion was unfortunately not measured in the present study.

Round 2

Reviewer 2 Report

Thanks you for the clarifications and adjustments provided.

One minor issue remains:

What does "improper NMES use" refer to (p2, section 2.1)?

Author Response

The patients used the NMES device also on the quadriceps of the operative side from wk 8 to wk 24. This was not permitted as NMES only had to be applied on the nonoperative side from wk 2 to wk 8. We realized the problem only during the last testing session. This has now been clarified.

Please see page 2, section 2.1.: "Two patients from the sham-NMES group discontinued the intervention and one patient from the NMES group was excluded from the analyses due to improper NMES use (NMES applied to both sides), resulting in a sample size of 12 patients per group."